# The Influence of Manganese Addition on the Properties of Biodegradable Zinc-Manganese-Calcium Alloys

**DOI:** 10.3390/ma16134655

**Published:** 2023-06-28

**Authors:** Wanda Mamrilla, Zuzana Molčanová, Beáta Ballóková, Miroslav Džupon, Róbert Džunda, Dávid Csík, Štefan Michalik, Maksym Lisnichuk, Karel Saksl

**Affiliations:** 1Department of Biomedical Engineering and Measurement, Faculty of Mechanical Engineering, Technical University of Košice, Letná 9, 040 01 Košice, Slovakia; wanda.mamrilla@tuke.sk; 2Slovak Academic of Science, Institute of Materials Research, Watsonova 47, 040 01 Košice, Slovakia; molcanova@saske.sk (Z.M.); bballokova@saske.sk (B.B.); mdzupon@saske.sk (M.D.); rdzunda@saske.sk (R.D.); dcsik@saske.sk (D.C.); 3Faculty of Materials, Metallurgy and Recycling, Institute of Materials and Quality, Technical University of Košice, Letná 9, 042 00 Košice, Slovakia; 4Diamond Light Source Ltd., Harwell Science and Innovation Campus, Didcot OX11 0DE, UK; stefan.michalik@diamond.ac.uk; 5Faculty of Science, Institute of Physics, Pavol Jozef Šafárik University in Košice, Park Angelinum 9, 040 01 Košice, Slovakia; maksym.lisnichuk@upjs.sk

**Keywords:** zinc-based alloys, microstructure, mechanical properties, hot extrusion, synchrotron data, EDX analysis

## Abstract

This study focuses on the preparation and characterization of zinc-based alloys containing magnesium, calcium, and manganese. The alloys were prepared by the melting of pure elements, casting them into graphite molds, and thermo-mechanically treating them via hot extrusion. The phase compositions of the samples were analyzed using X-ray diffraction technique and SEM/EDX analysis. The analysis confirmed that in addition to the Zn matrix, the materials are reinforced by the CaZn_13_, MgZn_2,_ and Mn-based precipitates. The mechanical properties of the alloys were ascertained by tensile, compressive, and bending tests, measurement of the samples microhardness and elastic modulus. The results indicate that an increase in Mn content leads to an increase in the maximum stress experienced under both tension and compression. However, the plastic deformation of the alloys decreases with increasing Mn content. This study provides valuable insights into the microstructural changes and mechanical behavior of zinc-based alloys containing magnesium, calcium, and manganese, which can be used to design alloys for specific biomedical applications.

## 1. Introduction

Biodegradable metals and alloys are materials that have the ability to degrade and dissolve over time in a biological environment. These materials offer several advantages for various applications, particularly in the field of biomedical engineering. They can be used for temporary implants, providing support, and/or treatment to the affected area while gradually degrading and being replaced by natural tissue. This eliminates the need for additional surgeries to remove permanent implants, reducing complications and promoting faster healing [1,2]. The alloys currently under investigation are predominantly comprised of iron (Fe), magnesium (Mg), and zinc (Zn) [3]. Although Fe alloys have good mechanical properties, their large elastic moduli can result in a stress-shielding effect. Additionally, their relatively slow degradation rates may result in the accumulation of corrosion products within the body, which could potentially trigger inflammation [4]. Mg alloys possess low densities and high specific strengths. However, their rapid degradation rates and release of hydrogen during the degradation process impose limitations on their potential applications [5,6]. Zinc alloys, on the other hand, exhibit a moderate corrosion potential, falling between those of iron and magnesium (Mg) alloys, which makes their degradation rates comparatively more suitable for certain applications [7]. Hence, Zn alloys are recognized as an advanced biodegradable material for the upcoming intracorporeal implants. Nevertheless, the mechanical properties of pure Zn are unsatisfactory and their enhancement can be achieved through effective alloying and deformation processing [8,9]. Previous studies have indicated that magnesium is an exceptionally effective element for alloying with Zn alloys, greatly improving their mechanical performance [10]. Calcium similarly enhances the mechanical properties of the alloys, making them stronger and more durable. Calcium also promotes bone regeneration and healing, facilitating better integration with the surrounding bone tissue and improving the overall performance of implants. Additionally, calcium plays a crucial role in the regulation of cellular processes, including cell adhesion and proliferation, which are important for tissue growth and healing. Manganese refines the grain size enhancing the strength of the biodegradable zinc alloys. Furthermore, manganese plays a role in influencing the corrosion rate of these alloys. It helps mitigate the corrosion process and extend the lifespan of the implants. This corrosion control is essential for ensuring the long-term stability and reliability of the zinc biodegradable alloys within the human body metabolism of amino acids and carbohydrates [11].

Multiple criteria have been established for materials utilized for bone implants, including the requirement for an ultimate tensile strength (UTS) between 200 and 300 MPa, an elongation at fracture of 15 to 20%, and a elastic modulus (E) falling within the range of 10 to 30 GPa [12]. Venezuela et al. proposed target values for orthopedic internal fixation devices, which include a yield strength (YS) greater than 230 MPa, an ultimate tensile strength (UTS) greater than 300 MPa, a elastic modulus (E) ranging from 10 to 20 GPa, and an elongation at fracture (ε) of 15 to 18% [13].

In a study conducted by Yang et al., Zn-Mg-Ca ternary alloys were investigated. The addition of Mg had the most significant positive impact on the mechanical properties of the alloys. The precipitation of the intermetallic phase, Mg_2_Zn_11_, resulted in a desirable hardening effect through dispersion strengthening. Among the studied alloys, ZnMg_1.5_Ca_0.1_ demonstrated the highest ultimate tensile strength of 442 MPa after undergoing hot extrusion. On the other hand, the ZnMg_1_Ca_0.1_ alloy exhibited the highest elongation of ε = 5.44% [14].

Liu et al. investigated Zn-1Mg-0.1Mn alloys and observed exceptional mechanical performance after hot rolling [15]. These alloys exhibited notable improvements, such as yield strength of 195 MPa, an ultimate tensile strength of 299 MPa, an elongation of 26%, and a hardness of 107.82 Hv.

System Zn3Mg0.7Y was studied by Panaghie et al. preparing by induction casting. Materials performed increased Young modulus 15.07, materials stiffness 1.65 and micorhardness 1.10 compared to pure Zn which achieved values of 2.9, 0.82 and 0.17, respectively [16].

Zhang-Zhi et al. aimed to improve mechanical properties by adding 0.4 wt% of Ag, Cu or Ca to biodegradable Zn-0.8Mn alloy. The ductility and the tensile strength were improved after extrusion. The YS, UTS and ε for Zn-0.8Mn-0.4Ag were 156 MPa, 251 MPa and 62%, respectively. For Zn-0.8Mn-0.4Cu alloy it was 191 MPa, 308 MPa and 39%, respectively. The highest values of YS 253 MPa and UTS 343 MPa are obtained for Zn-0.8Mn-0.4Ca alloy with ε of 8% [17].

This paper focuses on examining the impact of Mn on the microstructure and mechanical properties of the Zn-Mg-Ca ternary alloys. The objective is to establish a correlation between the chemical composition, microstructure, and mechanical properties of these alloys, with the aim of providing valuable data to optimize the composition design and processing of Zn-based biodegradable alloys containing Mn. It is a so far unstudied new system consisting of four biocompatible elements. The mechanical properties are improved not only by addition of Mn but laso by the processing technique.

## 2. Materials and Methods

### 2.1. Preparation of Alloys

The melt-stirring technique was used to fabricate experimental material. The raw metal materials were melted from high-purity zinc (Heneken 99.995%), magnesium (Alfa Aesar 99.98%), calcium (Alfa Aesar 99.5%), and manganese (Alfa Aesar 99.3%). Melting was performed in electric furnace at 650 °C under protective atmosphere of argon gas. The alloys were cast into a preheated graphite cylinder-shaped mold using the gravitational casting method. The as-cast ingots in the next step underwent a hot extrusion at temperature of 250 °C. The extrusion was carried out with an extrusion ratio of 21:1 and a speed of 3 mm/s. The resulting extruded rods had a circular cross-section with a diameter of 4 mm, and they were naturally cooled to room temperature in calm air. Each prepared alloy composition had a constant weight percentage of 0.4% for both Mg and Ca, while the weight percentage of Mn varied from 0 to 1.1 wt.%. The chemical compositions of prepared alloys were determined by EDX analyses (Table 1). As from the table can be seen the composition of all the prepared alloys exhibited a difference of less than 0.07 wt.% compared to their nominal composition.

### 2.2. Microstructure and Phase Characterization

Metallographic analyses were performed in order to evaluate uniformity of the microstructure and to ascertain the average size of matrix grains. The specimens were first embedded in an electrically conductive material (PolyFast, Struers), followed by grinding with SiC (P-800) abrasive papers. In the final step, was polishing using a diamond abrasive paste containing particles of size up to 0.25 µm. The microstructure of the alloys was revealed by etching of the samples polished surface in the Nital etchant (Ethanol 96% and HNO_3_ 4%). The microstructure of both the as-cast and extruded specimens was examined using two different techniques. The first technique involved the use of the Olympus GX 71 optical microscope, which provides a macroscopic view of the sample to study its overall grain structure and phase distribution. For more detailed analysis, a scanning electron microscope (TESCAN VEGA-3 LMU, TESCAN Brno, s.r.o., Brno, Czech Republic) equipped by EDX spectrometer (Bruker Nano Xflash 410M, Billerica, MA, USA) was employed. The phase composition of the hot extruded samples was determined by analyzing X-ray diffraction (XRD) patterns. These XRD patterns were obtained from synchrotron measurements conducted at the I12-JEEP beamline [18] located at the Diamond Light Source in the Harwell Science and Innovation Campus in Oxfordshire, UK. The X-ray diffraction (XRD) measurements were conducted by irradiating the samples with an X-ray beam of cross-section 0.5 mm × 0.5 mm and photon energy of 108.69 keV. Diffraction images were collected in transmission geometry employing a 2M CdTe Pilatus detector situated in an asymmetric position in respect of an incoming X-ray beam. 2D diffraction images were azimuthally integrated into 1D intensity curves as a function of a scattering angle 2θ using the DAWN software, 2.26.0 [19].

Thin foils were prepared for detailed observation of the microstructure in a TEM. The experimental samples from hot extruded alloys were prepared by grinding/polishing and ion milling employing Ion Slicer EM-09100IS (JEOL, Tokyo, Japan). TEM images were acquired by means of High-resolution Scanning-transmission Electron Microscope (S)TEM JEOL 2100F (JEOL, Japan), equipped with a Schottky field emission gun, and operated at an acceleration voltage of 200 kV. For phase identification, the Selected Area Electron Diffraction (SAED) technique was used.

### 2.3. Mechanical Testing

The tensile strength and deformation properties of the alloys after extrusion were tested by the uniaxial tensile strength test. For the experiment, specimens of diameter 3 mm and a length of 15 mm were prepared according to the ASTM-E8-E8M standard [20]. The compressive strength was evaluated on cylindrical specimens with a diameter of ϕ = 4 mm and a length of 8 mm, and for the bending test, ϕ = 4 mm × 55 mm according to the ASTM E9-89 standard [21]. All mechanical properties were evaluated at room temperature by using a universal material testing machine (TIRATEST 2300, TIRA, Schalkau, Germany). The crosshead speed used for the tensile test was 0.1 mm/min, 0.2 mm/min for the compressive test, and 1 mm/min for the bending test. Each composition underwent a minimum of five tests to ensure the reproducibility of the results. After, the tests fracture surfaces of the specimens were examined using the Jeol JSM 7000F scanning electron microscopy. Microhardness measurements were performed on the hot extruded samples using a Wilson-Wolper Tukon 1102 microhardness tester with a Vickers indenter. The Vickers hardness was determined according to ISO 6507 [22]. The applied load was 0.1 kg with a dwelling time of 10 s.

## 3. Results

### 3.1. Microstructure of the Zn-Mg-Ca and Zn-Mg-Ca-Mn Alloys

The microstructure of the as-cast state experimental materials was examined, revealing polyhedral grains with an approximate size of 40 μm. After hot extrusion, the grains reduce to approximately 10 μm. The hot extrusion process allows grain refinement due to ongoing dynamic recrystallization, as well as fragmentation of the eutectic structure. Figure 1a,b shows the microstructures of the Zn-0.4Mg-0.4Ca alloy in the as-cast state and after extrusion, respectively. Figure 1c,d display the microstructures of the Zn-0.4Mg-0.4Ca-0.8Mn samples. Notably, rectangular precipitates are visible in the Mn-containing samples after deformation.

The EDX analysis conducted on the microstructure revealed the presence of various phases in the alloys, including Zn, CaZn_13_, MgZn_2_, and Mn. Figure 2a depicts the microstructure of the as-cast alloy, which consists of primary grains and eutectic structures at their boundaries. Primary grains form during the solidification process, while eutectic structures arise due to the eutectic reaction between solid phases during the cooling process. Irregular precipitates observed at the boundaries of the primary grains in Figure 2b,c were confirmed. The EDX analysis suggests presence of the MgZn_2_, CaZn_13_, and Mn-containing precipitates.

The synchrotron X-ray diffraction patterns provided confirmation of the presence of the mentioned phases in the alloy. The Zn matrix phase (ICDD: 00-004-0831 [23]) exhibited a hexagonal structure with the space group P63/mmc. Similarly, the MgZn_2_ phase (ICDD: 00-077-1177 [24]) also displayed a hexagonal structure with the same space group. The CaZn_13_ phase (ICDD: 00-028-0258 [25]) demonstrated a cubic structure with the space group Fm3c, as depicted in Figure 3. These structural characterizations contribute to our understanding of the alloy’s crystallographic properties and aid in further analysis of its behavior and performance.

Rietveld refinement of the synchrotron XRD data provided additional valuable information about the unit cell volumes of each phase (Figure 4a) and quantitative fraction of these phases, see Figure 4b).

Figure 5b shows TEM bright field image of the Zn-0.4Mg-0.4Ca sample. The TEM analysis confirms presence of the above-mentioned intermetallic phases. The larger grains were identified as the pure Zn phase with a space group of P63/mmc, while the smaller grains were identified as the CaZn_13_ phase with a space group of Fm-3c.

### 3.2. Mechanical Properties of Zn-Mg-Ca and Zn-Mg-Ca-Mn Alloys

Tensile, compressive, and three-point bending tests, as well as measurements of elastic moduli and microhardness, were performed in order to evaluate the influence of adding manganese to the Zn-Mg-Ca alloys. Based on the performed tests, it was confirmed that there is a direct correlation between the content of manganese and the maximum stresses in both tension and compression. As the Mn content increased, the maximum stresses observed during both tensile and compressive testing also increased. This indicates that the addition of Mn increases mechanical strength of the Zn-Mg-Ca alloys, resulting in higher stress tolerance in both tension and compression (Table 2). Among the tested alloys, the Zn-0.4Mg-0.4Ca-1.1Mn alloy exhibited the highest strength in tension having a UTS (ultimate tensile strength) of 379 MPa (Table 2) and a YS (yield strength) of 299 MPa, which are considerably higher values compared the pure Zn, UTS and YS of 30 MPa and 23 MPa, respectively [15]. However, the Zn-0.4Mg-0.4Ca-1.1Mn alloy exhibited the lowest elongation to fracture among the all the tested samples, only 1.6%. Despite its high tensile and yield strength, this alloy displayed reduced ductility, indicating a limited ability to undergo plastic deformation before fracture. The elastic moduli of the alloys were compared to the elastic modulus range of human bone, having elastic modulus between 5 and 23 GPa [26]. The results showed that the alloys had significantly higher values compared to human bone, Table 2. This suggests that the tested alloys possess a greater stiffness or rigidity than human bone, which may have implications for their use in load-bearing applications where matching or mimicking the mechanical properties of bone is desired. In compressive testing, the Zn-0.4Mg-0.4Ca-1.1Mn alloy demonstrated the highest UCS (ultimate compressive strength), reaching a maximum value of 645 MPa (Table 2). This indicates a significant improvement in the ultimate strength compared to pure zinc, which exhibited a value of 103 MPa [27]. Regarding mechanical stiffness, the alloys meet the requirements to be considered as potential candidates for load-bearing biodegradable implants and stents [28]. The stress–strain curves of tensile and compression tests are shown in Figure 6a,b.

In the three-point bending test, Zn-0.4Mg-0.4Ca-1.1Mn alloy exhibited the highest UBS (ultimate bending strength), with a maximum value of 162 MPa, which was 18.6% higher compared the Zn-0.4Mg-0.4Ca alloy. The stress–strain curves of the alloys showed that those with lower Mn concentrations had better ductility, whereas those with higher Mn concentrations became more brittle (Figure 6c).

Figure 7 and Table 3 present the results obtained from microhardness testing. The data clearly demonstrate that the microhardness (HV0.1) of the extruded materials increases with an increase in weight percentage of manganese. In general, the microhardness measurements indicate that the Zn-0.4Mg-0.4Ca-1.1Mn alloy exhibits a microhardness (HV0.1) that is 1.5 times higher compared to the Zn-0.4Mg-0.4Ca alloy (as shown in Figure 7). The observed increase in microhardness can be attributed to two factors: the formation of a harder solid solution matrix and the grain refinement caused by the presence of manganese.

### 3.3. Fractography Analysis

Fracture surface analysis was performed on the specimens used for the tensile tests, comparing alloys without Mn and with the maximum Mn content.

Figure 8a displays the fracture surface of the Zn-0.4Mg-0.4Ca alloy, while Figure 8b provides a closer view of a specific area on the fracture surface. The images clearly document that the fracture is primarily transcrystalline, characterized by cleavage planes, with limited evidence of plastic deformation. A macro image of the fracture surface documents a wrinkled surface which is a manifestation of some degree of macroplasticity of the material. The detailed image of the fracture surface reveals the presence of a network of parallel cracks that propagated within the grains. These cracks have originated at the grain boundaries. This indicates that the fracture process involved intergranular cracking, where cracks initiated and propagated along the interfaces between adjacent grains.

Figure 9a shows the fracture surface of the Zn-0.4Mg-0.4Ca-1.1Mn alloy, while Figure 9b provides a detailed view on it. The fine particle morphology observed in the eutectic mixture of this alloy indicates its ability to withstand higher loads before fracturing. This is attributed to the presence of Mn which enhances the strength and toughness of the material. The refined morphology of the eutectic mixture, particularly around the cleavage planes, suggests that the material underwent plastic deformation before fracturing. The presence of Mn in the alloy contributes to its enhanced mechanical properties, allowing it to withstand higher tensile, Table 2.

The correlation between the mixed fracture morphology and the mechanical properties of the examined alloys is indeed a crucial finding. The observation of thick and transcrystalline fractures in the Zn-0Mn alloy contrasts with the refined fracture surfaces and the presence of plastic deformation in the Zn-1.1Mn alloy having enhanced tensile strength. This correlation underscores the significance of alloy design in improving the mechanical properties of zinc-based materials. By carefully selecting and incorporating alloying elements, such as Mn, it is possible to enhance the strength and toughness of zinc-based alloys, resulting in improved mechanical performance. These findings provide valuable insights for alloy development and can guide the design of materials with desired mechanical properties.

## 4. Conclusions

The results of this experimental study can be summarized as follows:The melt-stirring combined with hot-extrusion proved to be a suitable technique for preparation of the Zn-Mg-Ca-Mn alloy.The addition of manganese was found to have a significant influence on both the microstructure and mechanical properties of the newly investigated biodegradable Zn-Mg-Ca-Mn material system.The microstructure analysis revealed a noticeable difference between the as-cast Zn materials showing polyhedric grains of diameter 40 μm and after extrusion with grains of size 10 μm. The weight fraction of manganese in alloy has a significant impact on both the mechanical properties and microstructure of the alloys.The microstructure and phase analysis confirms presence of the Zn, CaZn_13_, MgZn_2_, and Mn-containing phases in the eutectic structures of the alloys, with fraction varying depending on the composition.The addition of Mn resulted in an increase in the maximum ultimate strength (US) in both tension and compression, as well as an increase in the microhardness (HV0.1) of the materials. Among all those tested, the Zn-0.4Mg-0.4Ca-1.1Mn alloy exhibited the highest values for ultimate tensile strength (UTS) at 379 MPa, ultimate compressive strength (UCS) at 645 MPa, ultimate bending strength (UBS) at 162 MPa, and microhardness at 133 HV. These findings demonstrate the beneficial effect of Mn on enhancing the mechanical strength of the alloys. However, the incorporation of Mn also resulted in a decrease in ductility, as evidenced by the lowest elongation at fracture observed in the Zn-0.4Mg-0.4Ca-1.1Mn alloy.The results also showed that the mechanical properties of the alloys can be tailored by varying alloy composition and heat treatment, making them suitable for different applications. In conclusion, this study offers significant insights into the microstructural modifications and mechanical performance of zinc-based alloys. These findings serve as a valuable resource for guiding the development of novel alloys with enhanced properties.

## Figures and Tables

**Figure 1 materials-16-04655-f001:**
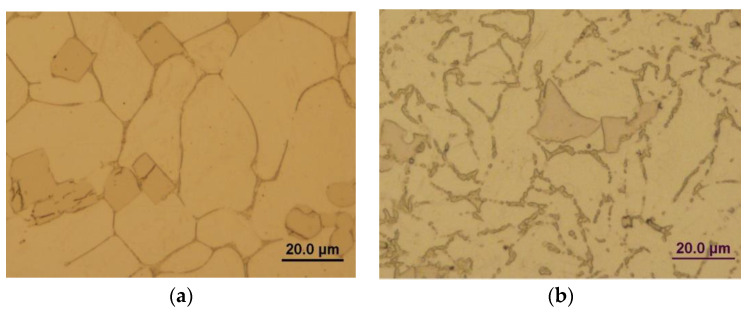
Optical micrographs of (**a**) Zn-0.4Mg-0.4Ca as-cast alloy, (**b**) Zn-0.4Mg-0.4Ca extruded alloy, (**c**) Zn-0.4Mg-0.4Ca-0.8Mn as-cast alloy, (**d**) Zn-0.4Mg-0.4Ca-0.8Mn extruded alloy.

**Figure 2 materials-16-04655-f002:**
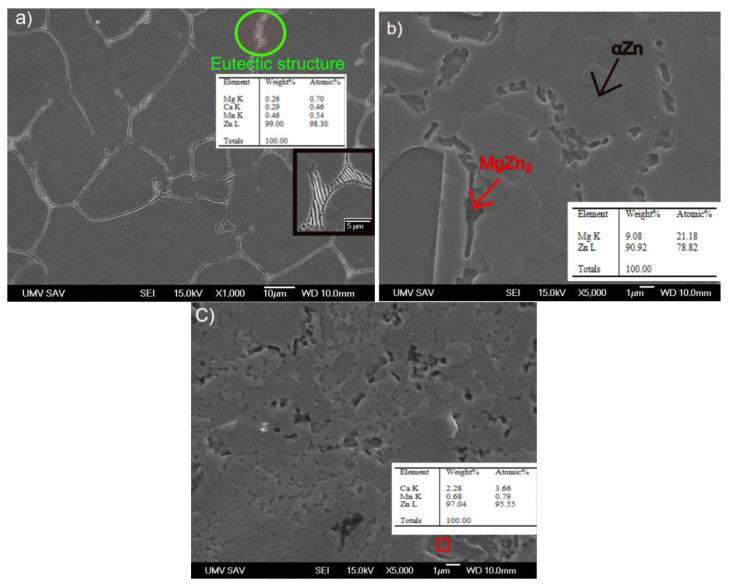
SEM micrographs (**a**) Zn-0.4Mg-0.4Ca-0.4Mn as-cast alloy with detail of eutectic structure, (**b**) Zn-0.4Mg-0.4Ca-0.4 Mn extruded alloy, (**c**) Zn-0.4Mg-0.4Ca-0.8Mn extruded alloy, red box represents region with precipitates.

**Figure 3 materials-16-04655-f003:**
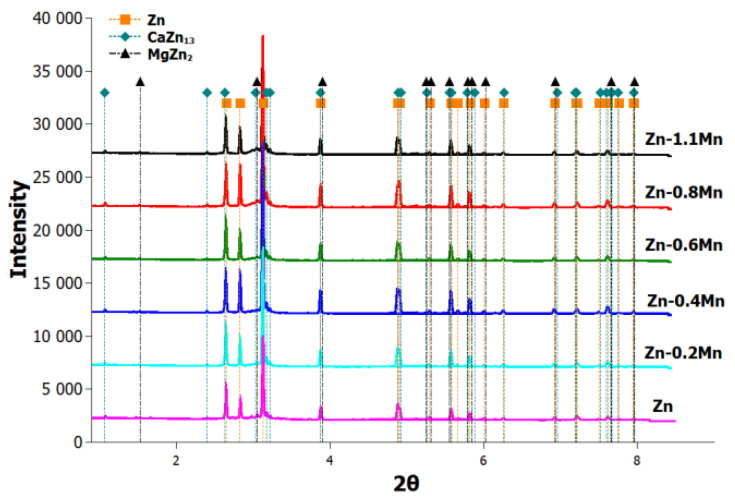
Synchrotron patterns of Zn-Mg-Ca-Mn extruded alloys, **Zn** represents Zn-0.4Mg-0.4Ca alloy.

**Figure 4 materials-16-04655-f004:**
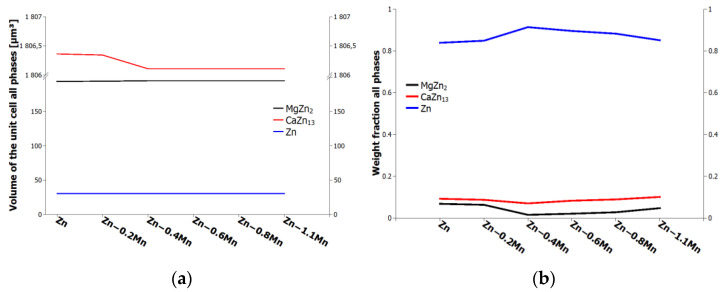
Synchrotron analysis: (**a**) volume of unit cell, (**b**) weight fraction all phases. **Zn** represents Zn-0.4Mg-0.4Ca alloy.

**Figure 5 materials-16-04655-f005:**
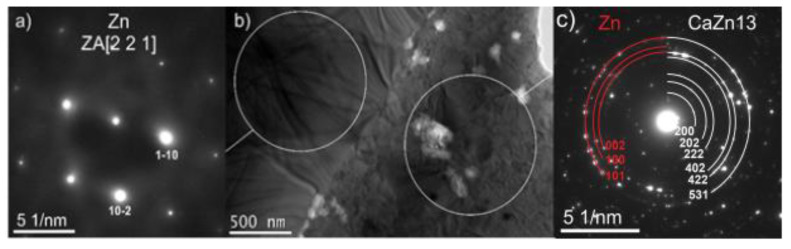
(**a**) SAED of the areas selected in the image (**b**), (**b**) TEM image of two different phases Zn and CaZn_13_ with magnification 10 kx, and (**c**) SAED of the areas selected in the image (**b**).

**Figure 6 materials-16-04655-f006:**
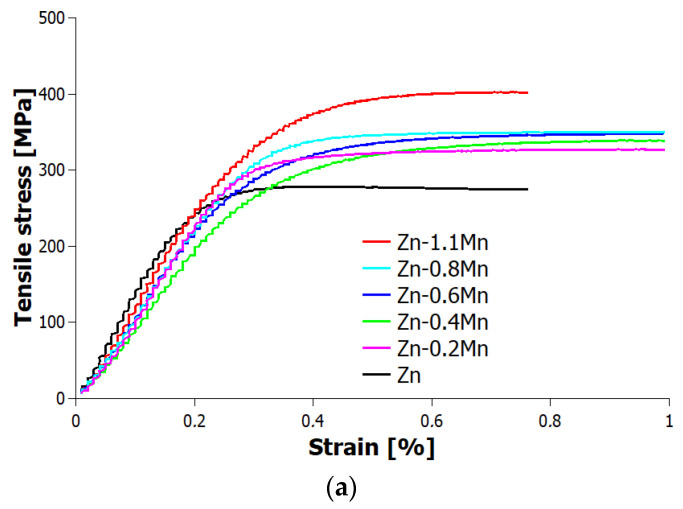
Ultimate strength and stress–strain curves of as-extruded alloys (**a**) tensile test, (**b**) compression test, and (**c**) three-point bending test.

**Figure 7 materials-16-04655-f007:**
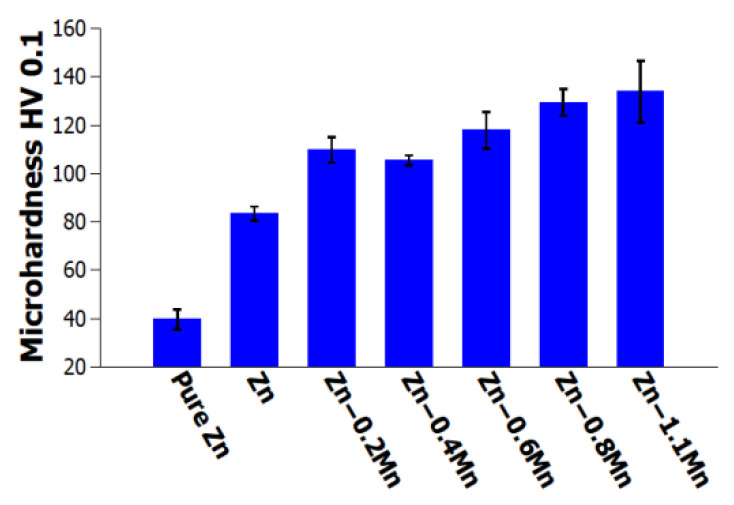
Microhardness HV0.1 of experimental materials.

**Figure 8 materials-16-04655-f008:**
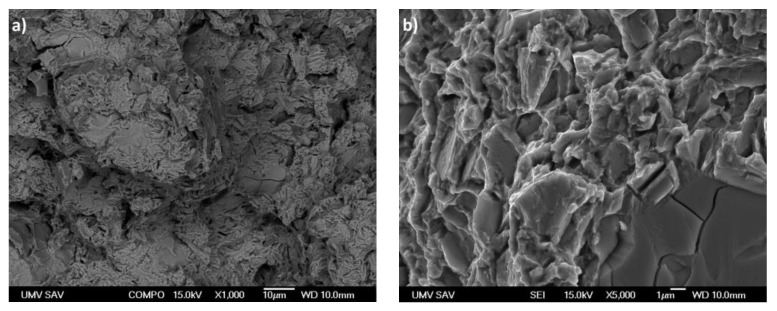
(**a**) Characteristic fracture surface of the alloy Zn-0.4Mg-0.4Ca after tensile test, (**b**) detail of the fracture surface.

**Figure 9 materials-16-04655-f009:**
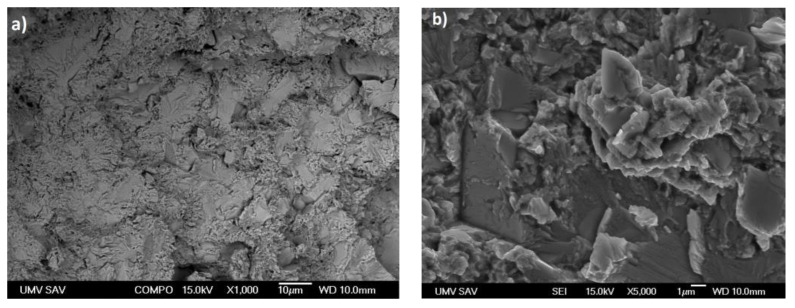
(**a**) Characteristic fracture surface of the alloy Zn-0.4Mg-0.4Ca-1.1Mn after tensile test, (**b**) detail of fracture surface.

**Table 1 materials-16-04655-t001:** Chemical composition of the prepared alloys.

Nominal Composition [wt%]		Actual Composition [wt%]		
	Zn	Mg	Ca	Mn
Zn-0.4Mg-0.4Ca	99.17	0.44	0.39	-
Zn-0.4Mg-0.4Ca-0.2Mn	99.05	0.41	0.39	0.15
Zn-0.4Mg-0.4Ca-0.4Mn	98.8	0.47	0.38	0.35
Zn-0.4Mg-0.4Ca-0.6Mn	98.58	0.44	0.43	0.55
Zn-0.4Mg-0.4Ca-0.8Mn	98.45	0.38	0.42	0.75
Zn-0.4Mg-0.4Ca-1.1Mn	98.04	0.48	0.38	1.1

**Table 2 materials-16-04655-t002:** Summarized mechanical properties of experimental materials.

As-Extruded Alloys	Label	Tensile Test				Compression Test		Bending Test
		YS [MPa]	σ_UTS_ [MPa]	ε [%]	E [GPa]	σ_UCS_ [MPa]	ε [%]	σ_UBS_ [MPa]
Pure Zn	Pure Zn	22.85 [15]	28	3.3 ± 0.1	96.5 [29]	102.92 ± 6.73 [27]	-	-
Zn-0.4Mg-0.4Ca	Zn-0Mn	253 ± 10	269 ± 6	4.2 ± 0.8	147 ± 6	486 ± 0	32 ± 1	137
Zn-0.4Mg-0.4Ca-0.2Mn	Zn-0.2Mn	277 ± 19	289 ± 14	2.7 ± 0.6	129 ± 2	558 ± 0	48 ± 1	140
Zn-0.4Mg-0.4Ca-0.4Mn	Zn-0.4Mn	289 ± 13	322 ± 7	3.6 ± 0.1	148 ± 2	566 ± 1	28 ± 1	140
Zn-0.4Mg-0.4Ca-0.6Mn	Zn-0.6Mn	287 ± 15	332 ± 7	3.5 ± 0.3	145 ± 3	626 ± 6	33 ± 1	153
Zn-0.4Mg-0.4Ca-0.8Mn	Zn-0.8Mn	275 ± 21	326 ± 8	2.5 ± 0.3	124 ± 3	616 ± 3	33 ± 1	159
Zn-0.4Mg-0.4Ca-1.1Mn	Zn-1.1Mn	299 ± 8	379 ± 2	1.6 ± 0.2	138 ± 6	645 ± 0	31	162

**Table 3 materials-16-04655-t003:** Microhardness HV0.1 of experimental materials.

As-Extruded Alloys	Microhardness HV0.1
Pure Zn	40 ± 2
Zn-0.4Mg-0.4Ca	83 ± 3
Zn-0.4Mg-0.4Ca-0.2Mn	110 ± 5
Zn-0.4Mg-0.4Ca-0.4Mn	105 ± 2
Zn-0.4Mg-0.4Ca-0.6Mn	118 ± 8
Zn-0.4Mg-0.4Ca-0.8Mn	129 ± 6
Zn-0.4Mg-0.4Ca-1.1Mn	134 ± 13

## Data Availability

Not applicable.

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
