# Peer review of "The Influence of Manganese Addition on the Properties of Biodegradable Zinc-Manganese-Calcium Alloys"

_materials, 2023, doi:10.3390/ma16134655_

Round 1

Reviewer 1 Report

In their work, Marilla et al. examined the roles of Mn alloying on the mechanical properties of biodegradable Zn-alloys. With the integration of microstructural characterization, mechanical testing, and fracture characteristics analyses, the authors showed that while Mn addition can lead to strength promotion, somewhat ductility loss occurs with high-Mn content. The research was nicely conceived with systematic experimental investigations and the major conclusion is rationally presented. Some of the observations, although not too surprising for in the Zn-alloy design, are still worth publishing to guide future research. I would like to support the publication of the present work, with the following minor comments/suggestions.

1. Because of the high vapor pressure, Zn can easily evaporate during the melting-casting process. The authors should then more clearly denote the error range and uncertainty in their composition measurements in Table 1.

2. Scale bars in Fig. 1 are all too small.

3. Please provide an enlarged micrograph to present the eutectic structure.

4. In the selected diffraction micrographs in Fig. 5 (c), there exists an extra spot between the 422 and the 531 rings. Which phase does such a diffraction signal correspond to? Please clarify.

5. In all the tensile curves in Fig. 6 (b), the elastic portions exhibit clear step-like serrations. The authors should clarify the origins or the possible artifacts that were involved in their measurements.

Author Response

  1. The uncertainty in the composition measurements are up to 0.07% as it is mentioned in the paper
  2. Figures were adjusted according to reviewer
  3. Figures were adjusted according to reviewer
  4. Diffraction signal correspond to phase Zn (100) : Figures were adjusted according to reviewer
  5. Step-like serrations are determined by the construction of the force and deformation sensors, and also by the method of storing data in the registers

Reviewer 2 Report

1)    Please incorporate the novelty of the work at the end of Introduction section.

2)    Please further strengthen the Introduction section by incorporating the recent literatures in the field of study, especially other metallic additions like, Cu, Mg etc. and the challenges incurred.

3)    Please provide the justification (May be literature support) for process parameters selection for preparation of alloys (section 2.1).

4)    Table 1 mentions about the fixed composition. The nominal composition is generally having composition range. What is the permissible composition variation in each alloying addition in the defined alloys? Please clarify.

5)    Please include the Microhardness measurement standard using Vickers hardness tester in the revised manuscript.

6)    Since it is hot extrusion is at high temperature (recrystallisation temperature), how grain refinement is happened (Figure 1)? Please explain and add the content in the revised manuscript.

7)    What is the sample size (number) for mechanical property analysis? Is it only one sample or the average? Please explain.

8)    Will the property obtained upon hot rolling will deteriorate due to natural aging at room temperature during the application (overaging phenomenon) after hot working by the agglomeration of intermetallics particles formed? Please comment.

9)    In the reference, some places pp. is inserted for page number. In majority of the cases pp. is missing. Please format uniformly.

10)                        The title can be modified as “The influence of Mn addition on the mechanical properties of biodegradable Zn-Mg-Ca alloys”. The property concentrated is only Mechanical.

Minor

Author Response

  1. Please see the attachment : Introduction section
  2. Please see the attachment : Introduction section
  3. Based on previous experience with the preparation of biodegradable materials, these preparation parameters were selected
  4. Maximum permissible composition variation  is 0.2%
  5. Please see the attachment : 2.3. Mechanical testing
  6. Please see the attachment :  3.1. 
  7.  Each mechanical test consisted of testing 5 samples for each composition, as it is mentioned in section 2.3. Mechanical testing. Results are evaluated as an average.
  8.  In this part of the presented research, we did not analyze the aging of the material. The paper is aimed at showing the improvement of mechanical properties. 
  9. Please see the attachment: References

Reviewer 3 Report

This manuscript investigated the synthesis and characterization of zinc-based alloys incorporating magnesium, calcium, and manganese. The alloys were prepared by melting pure elements and casting them into graphite molds. Thermo-mechanical treatment was applied through hot extrusion. The phase compositions of the samples were examined using X-ray diffraction and SEM/EDX analysis. These analyses confirmed the presence of reinforcing precipitates such as CaZn13, MgZn2, and manganese-based compounds in addition to the zinc matrix.

To evaluate the mechanical properties of the alloys, the authors have conducted various tests including tensile, compressive, and bending tests, as well as measurements of microhardness and elastic modulus. The authors have shown an increase in manganese content led to higher maximum stress levels under both tension and compression. However, the plastic deformation of the alloys decreased with higher manganese content.

This study offers valuable insights into the microstructural changes and mechanical behavior of zinc-based alloys containing magnesium, calcium, and manganese. The findings can be utilized to design alloys tailored for specific biomedical applications. The manuscript is well written and organized and it may have some originality. However, the following list provides some suggestions for major revisions which may help improve the manuscript:

1.            The experimental section is not clear on metallography procedures including proper referencing using ASTM and/or ISO standard procedures for grinding process of specimens and the sandpapers’ grit size and the applied force during each step.

Author Response

1. metallography procedures  were performed using ASTM E3-11(2017) and sandpapers’ grit size was P-800 as it is mentioned in the paper.

Reviewer 4 Report

The paper "The influence of Mn addition on the properties of biodegradable Zn-Mg-Ca alloys" is suitable for publication in Materials Journal and highlights very important aspects regarding Zn-based biodegradable alloys. The manuscript can be accepted after some minor updates.

1. The introduction is a little bit short; the authors should introduce aspects regarding the final use and applications of Zn-containing alloys. Also in comparison with other types of biomaterials, like titanium or Co-Cr alloys. Suggested references: 10.3390/ma16062487, 10.3390/ma14226806,

 10.24425/amm.2022.137794.

2. In Figure 1, please highlight the specific Zn compounds for the geometrical forms.

3. In the XRD description, please add the ICDD files and more discussion of the peak variation between alloys.

4.Reference list is short. Please update it.

The rest is fine.

Author Response

  1. Thematically, our paper does not deal with permanent Ti-based metal materials for tissue replacement or electrochemical testing. We included third of the suggested references in the introduction.
  2.  EDX analysis showed that lighter phase coresponds to Zn matrix and darker phases are CaZn13 and MgZn2
  3. We have added the ICDD number of the identified phases. The relative intensity of the Bragg´s peaks of all samples are similar while their volume fractions are shown on figure 4b
